# Auditory Electrophysiology of an Adolescent with Both Language and Learning Disorders

**DOI:** 10.3390/diagnostics15212779

**Published:** 2025-11-02

**Authors:** Aparecido J. Couto Soares, Adriana Neves de Andrade, Piotr Henryk Skarzynki, Claudia Berlim de Mello, Milaine Dominici Sanfins

**Affiliations:** 1Department of Speech-Hearing-Language, Universidade Federal de São Paulo, São Paulo 04044-020, SP, Brazil; 2Department of Teleaudiology and Screening, World Hearing Center, Institute of Physiology and Pathology of Hearing, 05-830 Kajetany, Poland; 3Center of Hearing and Speech Medincus, 05-830 Kajetany, Poland; 4 Institute of Sensory Organs, 05-830 Warsaw, Poland; 5Department of Psychobiology, Universidade Federal de São Paulo, São Paulo 04724-000, SP, Brazil; 6Post-Graduate Program in Clinical Audiology, Instituto de Ensino e Pesquisa Albert Einstein, São Paulo 05652-000, SP, Brazil

**Keywords:** specific language impairment, comorbidity, auditory evoked potentials, hearing

## Abstract

**Background and Clinical Significance:** developmental language disorder (DLD) and specific learning disorder (SLD) may coexist, resulting in significantly broader impairments to oral and written language skills. Understanding the neurobiological basis of these deficits is crucial, and electrophysiological assessment of the auditory system offers an objective approach not influenced by behavioral factors. The present study describes the audiological electrophysiology of an adolescent diagnosed with both DLD and SLD. **Case Presentation:** R.B., a 15-year-old adolescent with a history of SLD and DLD, presented with persistent deficits in oral language (syntax) and written (decoding) skills after 7 months of intensive therapy. Basic audiological tests confirmed hearing within normal limits. An electrophysiological battery, including the click-brainstem auditory evoked potential (c-ABR), medium latency auditory evoked potential (MLAEP), long-latency auditory evoked potential (P300), and frequency following response (FFR), was performed to investigate information processing in the auditory trajectory. The c-ABR confirmed the integrity of the auditory pathway up to the brainstem. MLAEP revealed a differential ear effect, with significant asymmetry in the Na-Pa interamplitude, pointing to a dysfunction in subcortical processing. The P300 showed a prolonged latency in the left ear (437 ms), and there was no response in the right. The FFRs under ideal and impaired listening conditions demonstrated impaired perception of speech and revealed that the neurophysiological responses did not correspond to the eliciting stimulus. **Conclusions:** The present case study showed that electrophysiological testing of the auditory system provided objective and quantitative evidence for a neurobiological basis of the language deficits of an adolescent with DLD and SLD. The work demonstrated that when comorbidities are present, a multidisciplinary investigation of both the linguistic and auditory systems can be helpful.

## 1. Introduction

Specific learning disorder (SLD) is a neurodevelopmental disorder that affects the way various academic skills are learnt. Some individuals with SLD may have large deficits in reading words accurately and fluently. There may also be losses in writing, including grammar, spelling, and letter shape. Impairments in numeracy, memorization of arithmetic facts, and calculation skills can also be observed [1].

In contrast, developmental language disorder (DLD) refers to cases of linguistic impairment in the absence of a known etiological condition, as described by Bishop et al. (2017) [2]. DLD involves impairments in language subsystems, phonological memory, and discursive skills [2]. It also involves changes in metaphonological skills and in the acquisition of written language [3,4]. DLD and SLD can both occur together [4,5].

SLD and DLD significantly alter auditory processes, both behaviorally and electrophysiologically. According to Shafer et al. (2005) [6], children with DLD show compromised responses in mismatch negativity potentials (MMNs), which is perhaps an indication that the children have somehow misidentified or wrongly processed speech information. Another study by Soares and colleagues reported that individuals with reading and writing disorders performed below normal on central auditory processing (CAP) tests. Additionally, these same subjects appeared to have a distinct pattern of MMNs. Soares et al. also found indications of an association between CAP results and features of long-latency auditory evoked potentials (LLAEPs) [7].

LLAEPs are a non-invasive way of probing neuroelectrical activity in the cortex in response to a sound stimulus [8]. Evaluation of one event-related potential (ERP), the P300, makes it possible to analyze the way information is processed sequentially in the cortex, giving insights into short-term memory, concentration, and decision-making skills, which are key aspects of speaking, language, learning, and writing [7,8,9,10]. Learning requires all these skills, so assessing auditory function in subjects who have impaired language function provides the opportunity to uncover important scientific insights.

Other neurophysiological techniques, such as the frequency following response (FFR), also provide avenues for investigating the ways in which speech sounds in the auditory pathway are transmitted. Speech sounds need to be first coded in the periphery and then decoded in the brain before acoustic, semantic, and linguistic information can be extracted, so understanding the stages in these processes reveals additional valuable information about how speech sounds are coded and decoded [11,12,13,14]. Any electrophysiological changes may reflect impairments in peripheral or central areas of the auditory system [11,12,15]. The idea is that there could be a correlation between language skills and certain neurophysiological responses in the FFR, so that specific alterations could reflect cases of learning disorders [12,16], school difficulties [11], central auditory processing disorders, or autism spectrum disorder [17].

It is widely known that both for DLD and SLD therapy interventions focusing on language and metaphological aspects are considered effective to improve not just language and phonological skills. They are also important in improving some auditory skills, but the majority of studies separate these two conditions. Thus, at the moment, little is known about the electrophysiological profile of individuals who have a dual diagnosis of language and learning disorders. The present study aims to use electrophysiological tests to investigate the hearing pathways of an adolescent who had both DLD and SLD.

## 2. Detailed Case Description

This study was approved by the Research Ethics Committee under process No. 7.115.963. The guardians responsible for the research participants were informed about the study and consented to participate by signing a Free and Informed Consent Form. Data collection took place prospectively between the periods of April and July 2025. It is important to mention that the patient was evaluated within the Unified Health System (SUS system) in Brazil, which serves low-income individuals; for this reason, advanced imaging techniques could not be applied.

R.B. is a 15-year-old male, born at term, with no complications during delivery and an Apgar score of 10 in the first and fifth minutes of life. The family’s complaint involved impairment of oral language and, especially, learning. There were no complaints regarding neuropsychomotor development, and the otological history was good. He had a good relationship with his family and peers, and there were no behavioral complaints from his school.

R.B. was evaluated by means of standard language tests in relation to semantic subsystems [18,19], phonology [20], syntax [21], phonological awareness [22], phonological memory [23], and speed of naming [24]. Decoding skills were also evaluated [25], as were fluency and reading comprehension [26], encoding [27], spelling [27], and handwriting. The results indicated deficits in all areas. With the exception of the phonological profile and rapid naming, R.B. performed below normal in all the other tests, falling in the lowest one percent. Performance was also impaired in oral language, phonological processing, and written language skills. He underwent a neuropsychological evaluation, in which he showed normal verbal and non-verbal intellectual function, and this was also the case for the speed of information processing.

Following DSM-5 recommendations (APA, 2023) [1], R.B. entered weekly diagnostic therapy focusing on semantic, syntactic, and discursive oral language skills, and phonological memory, phonological awareness, and rapid automatic naming (RAM), as well as therapy for reading and writing. Therapy involved 28 weekly sessions over 7 months, each 50 min in length. In every session, all deficient skills were worked on. R.B. was assiduous and only missed one session.

At the end of this period, R.B. was reassessed in the same areas as before. Consistent improvement was observed in all of the areas he worked on, with significant gains in scores, but he still did not reach the expected scores for his age group. In terms of the syntactic component of oral language, R.B. remained at a very low score, classified in the lowest one percent. In fact, persistent morphosyntactic alteration is an important marker for the diagnosis of DLD.

Regarding phonological processing, there were consistent gains in RAN, and R.B. scored close to that expected for his age group. However, there were persistent alterations in phonological memory and phonological awareness, alterations that were consistent with what is typically observed when diagnosing DLD at school age. The consistent recovery of RAN after a period of directed intervention indicates a minimal deficit. With regard to written language, there were a few gains in decoding words and non-words, as well as in writing isolated words. Due to persistent deficits, it was not possible to assess the fluency and comprehension of text, nor of its written production.

Considering the responses to written language skills, we conclude, based on the literature, that his disorders did not occur in the way typical of those with isolated DLD. The persistent deficits in acquiring writing skills point to a diagnosis of SLD in co-occurrence with DLD.

## 3. Electrophysiological Hearing Assessment

The audiological battery was performed at the Audiology and Electrophysiology Outpatient Clinic of the Paulista School of Medicine, Federal University of São Paulo (UNIFESP), São Paulo, Brazil. Hearing thresholds were found to be within normal limits (≤20 dB from 250 to 8000 Hz) using an GSI Pello ^TM^ audiometer, GSI, Eden Prairie, MN, USA), and there was a normal speech recognition index (WHO, 2021 [28]). Type A tympanograms and the presence of ipsi and contralateral acoustic reflexes (Jerger, 1970) [29] were confirmed with the TympStar Pro 2 ^TM^ equipment (GSI, Eden Prairie, MN, USA).

Electrophysiological evaluations were performed in a sound-attenuating, electrically shielded room where the subject remained comfortably seated in a reclining chair. The following procedures were performed: click auditory brainstem responses (c-ABR), middle-latency auditory evoked potential (MLAEP), event-related potentials (ERPs), P300 potential, and frequency following response (FFR). Depending on the procedure performed, positioning of the electrodes followed the recommendations of the international 10–20 system (Jasper, 1958) [30]. During recording, impedance was kept below 5 kΩ and inter-electrode impedance below 3 kΩ (see Table 1).

In evaluation of the c-ABR, results showed a normal auditory pathway up to the brainstem in both ears [31] (see Figure 1).

In the MLAEP, an alteration in the ear effect type was observed, considering the C3M2 × C3M1 and C4M2 × C4M1 leads, with response asymmetry greater than 50% in the Na-Pa interamplitude values, and no asymmetry was identified in the Nb-Pb analysis based on normative values [32] (Figure 2).

In the ERP-P300, prolongation was observed in the latency of the P300 wave (436.8 ms) in the left ear, with 69.6% of the rare stimuli properly identified; the mean latency was 392 ms. In the right ear, however, the P300 wave was absent; the success rate was only 38.9% and the latency was 400 ms longer than the normative values for the age group [8] (Figure 3).

In the FFR, it was possible to observe differences in the latency values of waves A, F, and O, considering the evaluation conditions in silence and noise. However, the slope and area values showed differences (silence × noise), and the correlational analyses showed similarities in the values in the collections (silence × noise) (Figure 4).

## 4. Discussion

Although language development requires the use of different skills and systems, the auditory system is fundamental. In any differential diagnosis and understanding of language and learning disorders, testing the auditory system is the first step. Such steps allow the integrity and functioning of the full auditory pathways to be investigated.

Auditory evoked potentials allow the auditory system to be objectively evaluated, which is important in cases where there is some type of impairment to communication. If there are impairments to language, speech, and learning skills, can they be separated from auditory responses? If neurophysiological data are not affected by behavioral or even cognitive impairments, this would mean a more reliable assessment could be made [11].

The first step is to ensure that the patient has audibility within normal limits, and in this case, it was readily confirmed.

Next, evaluation of brainstem auditory evoked potentials using click stimuli allowed the integrity of the auditory pathway to the brainstem to be confirmed. These measures were needed before higher-level auditory areas could be investigated.

However, studies of auditory evoked potential in cases of learning disorders are controversial. The present study found an absence of alterations in the c-ABR, which is in agreement with most authors [11,33,34]. Nevertheless, even if alterations are not identified, testing should be regularly performed in order to establish a normal baseline [11,35]. As highlighted by Jiang et al. (1995) [36] and Bhutta et al. (2002) [37], when evaluating c-ABRs in cases of neurodevelopmental disorders, special attention should be paid to perinatal risk factors, since any complications can sometimes affect c-ABRs in a way that is not directly related to language.

To investigate subcortical auditory pathways, MLAEPs are important tools. This potential allows an early and reliable diagnosis in patients with learning, speech, and language disorders to be made, and is important in monitoring therapeutic interventions. In the present study, a clear asymmetry between the right and left ears was identified, and such an ear effect is often found in patients with learning disorders [38,39].

Asymmetry is calculated in terms of the interamplitude values of the Na-Pa and/or Nb-Pb components (i.e., the difference between Na amplitude versus Pa amplitude), and the Na-Pa measurement is highly recommended in individuals with a history of learning disorders or other abnormalities. The Na component derives from the activity of the medial geniculate body and the thalamic polysensorineural nucleus, while the Pa component originates from neuronal activity in the inferior colliculus, medial geniculate body, brain areas involving Heschl’s gyrus, and the superior temporal gyrus [40]. Successful processing of auditory information requires the effective functioning of these structures. The inferior colliculus analyzes different acoustic aspects, such as frequency and intensity, and Heschl’s gyrus discriminates and recognizes sounds. The medial geniculate body is not directly involved in linguistic processing but is essential for the comprehension of verbal and non-verbal sounds [40,41]. Finally, the superior temporal gyrus allows us to recognize and understand speech.

Poor processing of auditory information due to deficiencies in the coding and organization of sound stimuli in individuals with learning disorders is likely to give rise to alterations in MLAEP responses and, consequently, cause impairments in the semantic, phonological, syntactic, and linguistic subsystems. Coding, decoding, and phonological awareness skills can also be affected [42]. The changes we saw in MLAEP responses were in line with previous research [38,43,44,45,46].

In terms of gaining insights into the higher levels of the auditory pathway, event-related potentials (ERPs), including the P300 potential, play an important role. Electrophysiological assessments, therefore, provide some insight into mental and cognitive processes such as focus, attention, memory, and decision-making, which are in turn related to language skills [47,48,49].

The present study showed that R.B. had alterations in both ears. Upon consideration of the normative values, the absolute latency values of the N1 component are within the normal range in both ears. The P2 components’ absolute latency values are bilaterally prolonged in accordance with what is considered normative [8]. It is important to mention that the origin of the N1 component derives from the supratemporal auditory cortex, while the P2 component derives from the lateral-frontal supratemporal auditory cortex, and, therefore, they perform different functions within auditory processing. The N1 component seems to be related to the process of habituation and non-associative learning, while the P2 component is evidence of the neuroplasticity of the auditory system. In addition to that, the P2 component is modulated by attention and involves primary and association cortical areas, and is thus dependent on attention, memory, executive function, and language skills [8,9,38,39,40]. The prolonged N1/P2 relationship in the right ear may reflect a sensory dysfunction or even an overload in the processing of auditory information, which corroborates a failure in the sound habituation process and, therefore, greater difficulty in determining which stimulus is more relevant in the evaluation. Another relevant aspect is that the latency values for both the N1 and P2 components were higher in the right ear, which was precisely the ear without the presence of the P300 component. And in light of these results, it would be pertinent to emphasize that the right ear is dominant for processing linguistic aspects [2,4,5,7,8,11,47].

In the right ear, even with previous training and understanding of the task to be performed, he was not able to respond consistently, recognizing less than 40% of the rare stimuli.

In the left ear, the correct identification of rare stimuli reached 69.6%, and there was a clear P300 potential, although its latency (437 ms) was prolonged compared to normative values for his age [8]. In the present study, other factors associated with cognitive responses—mean response time and response deviation rate (RMS)—were analyzed. The average response time is related to the patient’s average reaction time to the rare sound stimulus, which was 392 ms, with a standard deviation of 57.6 ms. The increase in response time and standard deviation is associated with attentional difficulties in individuals with ADHD, schizophrenia, and older adults, where there is also a prolongation of latencies of the P300 cognitive potential [50]. However, there are still no studies on these measures in individuals with learning disorders.

Language alterations can also affect the generation of the P300 potential, since language generally involves lower information processing speeds. The prolonged latencies we observed in this subject are a reflection of the way he processes sound stimuli and how his memory affects information retrieval [8,9,10,47]. Prolongation in P300 latencies is often seen in cases of children with learning difficulties and a history of repeated classes.

Learning a language requires assimilating the acoustic and phonetic elements of a language, and evaluation of the frequency following response (FFR) makes it possible to study the path of a sound along the entire trajectory of the auditory pathway. The FFR can thus provide a detailed insight into the perception and processing of verbal sounds. In individuals with learning disorders, the FFR can be used as a marker of school, speech, and learning difficulties, and is the only procedure within an audiological battery that can show such alterations [11].

By analyzing the coding of speech sounds in an ideal listening condition (in quiet) and in a difficult listening condition (noise), another aspect of the auditory system, that of understanding speech, can be examined. The analyses on R.B. were performed in both the time and frequency domains, with the latter providing a deeper insight into the neurophysiological responses.

R.B.’s answers demonstrated that, even under ideal listening conditions, when the sound stimulus /ba/ was presented, there was a disparity between the stimulus that was heard and the neurophysiological response. The correlation in quiet was 0.10, and when noise was presented to the contralateral ear, the correlation decreased to 0.08. All data—the signal-to-noise, root mean square, and lag ratio—demonstrate that the responses were robust and well-defined. In the quiet-to-noise correlational analysis, the data showed that there was a correspondence between neuronal activity under both listening conditions. This included the lag measurements, which measured the conduction time for the sound stimulus to propagate along the auditory pathway, and these were similar in the ideal (quiet) and impaired (noise) listening conditions.

The abnormal responses found in the FFR assessment are related to the inability to read fluently, since fluency demands neural stability and integration within the central auditory nervous system [51,52]. Cognitive processing also requires effective neurophysiological activity, so this may explain the observed impairments in the P300 cognitive potential and the FFR. In the case of individuals with both SLD and DLD, we expect to see broad alterations in oral language (vocabulary, morphosyntax, and speech), consistent changes in phonological processing, usually more severe than observed in pure DLD or pure SLD (with changes including in RAN), consistent difficulties in decoding and reading fluency, and deficits in reading comprehension and vocabulary [4,53].

In view of the connections between auditory processing and language, it is important to research language and hearing in an integrated way, since treating them in isolation may fail to identify comorbidity.

The assessment of potentials shows disorders that are consistent with the deficits observed in oral and written language, indicating that it is a sensitive measure for better characterization of such impairments, even when they occur comorbidly. In addition, the data indicate the need for future studies that include auditory training in these cases in order to enhance therapeutic gains and verify whether there are changes in the electrophysiological profile. In this sense, knowing the auditory profile of an individual with both pathologies makes it possible to prepare programs to stimulate auditory abilities in order to enhance their intervention process. More studies are needed with larger samples in order to exactly delimit their auditory profile and, hence, an auditory program to provide better support for their intervention process.

## 5. Conclusions

The present case study has demonstrated that electrophysiological assessment of the auditory system can provide objective and quantitative evidence of a neurobiological basis for the deficits observed in an adolescent diagnosed with both DLD and SLD. The normal c-ABR responses showed that there were no problems with the auditory pathway up to the brainstem level, leaving the upper auditory pathways open for investigation. The MLAEP showed a response asymmetry and impairment of the P300 responses, revealing dysfunctions in subcortical and cognitive processing. In addition, the FFRs, under conditions of both quiet and noise, demonstrated that the process of perceiving speech sounds was impaired. All these neurophysiological abnormalities can present significant barriers to the acquisition and consolidation of language and literacy skills.

## Figures and Tables

**Figure 1 diagnostics-15-02779-f001:**
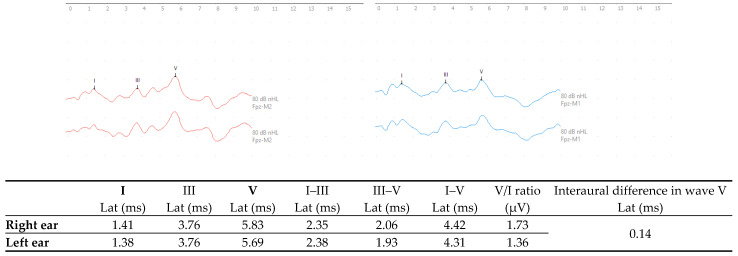
c-ABR responses with monaural click stimulation in the right ear (red) and left ear (blue) at 80 dBnHL. (Lat—latency; ms—milliseconds; μV—microvolts).

**Figure 2 diagnostics-15-02779-f002:**
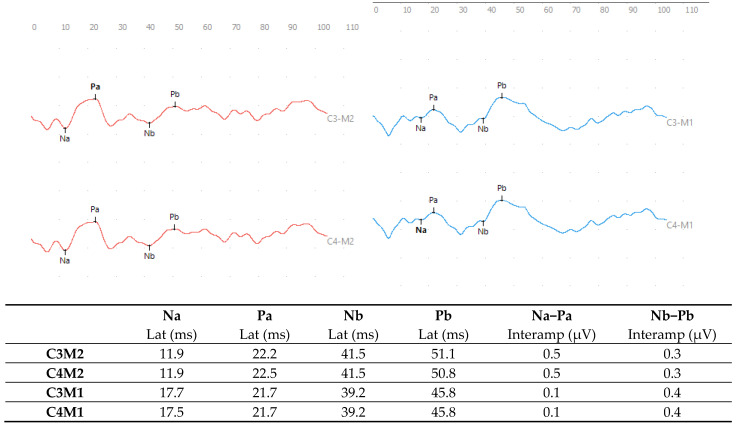
MLAEP responses with monaural click stimulation in the right ear (red) and left ear (blue) at 70 dBnHL.

**Figure 3 diagnostics-15-02779-f003:**
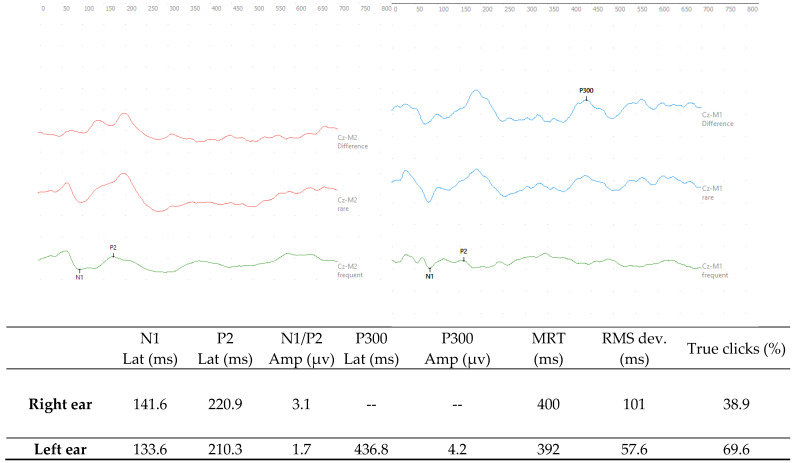
Responses of the ERP-P300 with monaural stimulation in the right and left ears, respectively, at 75 dBnHL with tone burst stimuli at the frequencies of 1000 Hz and 2000 Hz. Legend: Lat—Latency; Amp—Amplitude; ms -milliseconds; µv—microvolts; MRT—mean response time; and RMS—root mean square.

**Figure 4 diagnostics-15-02779-f004:**
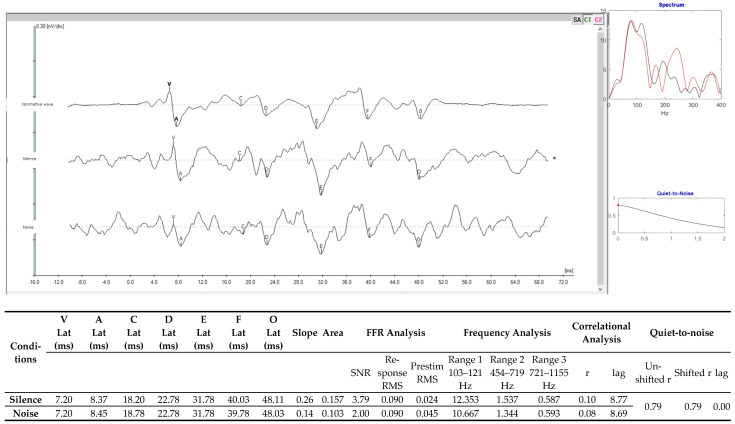
FFRs with the stimulation condition in silence and with contralateral noise in the right ear at 80 dB SPL with the speech stimulus /da/ of 40 ms. Legend: Lat—latency; ms—milliseconds; FFR—frequency following response; µv—microvolts; MRT—mean response time; RMS—root mean square; and SNR—signal-to-noise ratio.

**Table 1 diagnostics-15-02779-t001:** Parameters for auditory electrophysiology.

Procedure	Electrode Placement	Intensity	Collection Filter	Analysis Filter	Equipment	Transducer
c-ABR	M1, M2, Fz	80 dB HL	1–30 Hz	no filter	NeuroAudio^TM^—Neurosoft, Ivanovo, Rússia	Insert (ER-3A)
MLAEP	M1, M2, C3, C4, Fz	70 dB HL	20–1500 Hz	20–200 Hz	NeuroAudio^TM^—Neurosoft, Ivanovo, Rússia	Insert (ER-3A)
ERP-P300	M1, M2, Cz, Fz	75 dB HL	1–30 Hz	no filter	NeuroAudio^TM^—Neurosoft, Ivanovo, Rússia	Insert (ER-3A)
FFR	M1, M2, Cz	80 dB SPL	100–2000 Hz	no filter	Navigator Pro^TM^—Natus, Middleton, WI, USA	Insert (ER-3A)

## Data Availability

The raw data supporting the conclusions of this article will be made available by the authors upon request due to privacy concerns.

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
