# Peer review of "Auditory Electrophysiology of an Adolescent with Both Language and Learning Disorders"

_diagnostics, 2025, doi:10.3390/diagnostics15212779_

Round 1
Reviewer 1 Report
Comments and Suggestions for Authors
This research describes an auditory electrophysiological study that showed altered higher auditory sub-cortical and cortical level electrical activity in a patient with both DLD and SLD.
The background of the literature is provided, and the introduction is unambiguous. Although the example is clearly explained, the MLAEP results shown in the graphic are unclear. In the left ear stimulation, where is the Na component? The trace recording appears to be cut by the figure.
I want to watch every MLAEP recording.
The results and the discussion make sense, and the references are properly mentioned.
I would like to review the manuscript with the recommended correction.
Reviewer 2 Report
Comments and Suggestions for Authors
In the introduction section line 66, what does PAC means? Please explain.
In addition to the introduction section, since DLD and SLD are initially written out in full, it is considered acceptable to use abbreviations thereafter.
While the findings are understood, imaging evaluations such as CT, MRI, and SPECT are generally considered effective for understanding the underlying pathology. Was there a specific reason these assessments were not conducted in this case?
Finally, what intervention approaches are considered effective for children presenting with both SLI and DLD?
Reviewer 3 Report
Comments and Suggestions for Authors
The case report is the auditory electrophysiological description of a teenager with Developmental Language Disorder (DLD) and Specific Learning Disorder (SLD).
The primary pathology is well documented from a general point of view to familiarize the readers with a specialty different from neurology. Even the electrophysiologic tests are well described in terms of neural generators and function-related.
The case description, results, and conclusion are perfectly consistent, and the reference list is complete.
There is only a lack, namely, latencies and symmetries of N1/P2 in the cortical responses (Fig.3), which should be complemented by comments similar to those for other electrophysiological responses. If available, it is also interesting to consider the otoemissions to speculate about the efferent system. The last request is not mandatory to respect.
